# RLIE: Rule Generation with Logistic Regression, Iterative Refinement, and Evaluation for Large Language Models

## Abstract

Nowadays, Large Lange Models (LLMs) are able to propose rules in natural language, overcoming constrains of a predefined predicate space inherent in traditional rule learning. However, existing methods using LLMs often overlook the combination effects of rules, and the potential of coupling LLMs with probabilistic rule learning to ensure robust inference is not fully explored. To address this gap, we introduce **RLIE**, a unified framework that integrates LLMs with probabilistic modeling to learn a set of probabilistic rules. The RLIE framework comprises four stages: (1) **R**ule generation, where a LLM proposes and filters candidate rules; (2) **L**ogistic regression, which learns the probabilistic weights of the rules for global selection and calibration; (3) **I**terative refinement, which continuously optimizes the rule set based on prediction errors; and (4) **E**valuation, which compares the performance of the weighted rule set as a direct classifier against various methods of injecting the rules into an LLM. Generated rules are the evaluated with different inference strategies on multiple real-world datasets. While applying rules directly with corresponding weights brings us superior performance, prompting LLMs with rules, weights and classification results from the logistic model will surprising degrade the performance. This result aligns with the observation that LLMs excel at semantic generation and interpretation but are less reliable at fine-grained, controlled probabilistic integration. Our work investigates the potentials and limitations of using LLMs for inductive reasoning tasks, proposing a unified framework which integrates LLMs with classic probabilistic rule combination methods, paving the way for more reliable neuro-symbolic reasoning systems.

## 1 Introduction

In data-driven applications and scientific discovery, the goal is not merely to predict outcomes, but to construct a set of verifiable, reusable, and composable theories(Zhou et al., 2024; Yang et al., 2024a; Minh et al., 2022). These theories can enable explainable, auditable decisions while driving the discovery of new knowledge and underlying structures(Yang et al., 2023; 2024b). These theories can be expressed in formal, structural statements(Cohen et al., 1995; Cropper & Morel, 2021) or natural language hypotheses(Zhou et al., 2024), and they share a common characteristic: they are declarative, testable, and self-contained discriminative patterns that yield predictions verifiable by external evidence[1]. The challenge of how to learn and integrate a set of such collaborative patterns to both uncover regularities and advance knowledge accumulation has long been a central problem in machine learning and artificial intelligence(Bazgir et al., 2025).

Take the binary classification task of spam detection as an example. A set of rules might include (1) HasToken('win prize', $x$) $\wedge$ HasAttachment($x$); (2) DomainInBlacklist($x$) $\wedge$ HasShortLink($x$); and (3) TitleAllCaps($x$) $\wedge$ HasToken('urgent', $x$), all implying the head "is spam". These rules cover different facets of the problem, and must work in concert. A classic approach to use the rules to do reasoning is deterministic aggregation, where a positive classification is made, if any rule is satisfied(Fürnkranz & Kliegr, 2015). A probabilistic approach, in contrast, treats each rule's classification result as a binary indicator ($r_j \in \{0, 1\}$) and employs a statistical model, such as logistic regression, to learn a global set of weights for combining these signals into a unified prediction(Ruczinski et al., 2003; Friedman & Popescu, 2008). While the well-defined structure of rules offers significant advantages, traditional rule learning methods are constrained by a predefined predicate space, which limits their ability to handle unstructured data and open-ended semantics, thereby restricting their applicability in broader scenarios(Cerna & Cropper, 2024; Hocquette et al., 2024).

---

[1]In this paper, we do not distinguish between the terms "rule" and "hypothesis", and will use "rule" throughout the text for consistency.

The recent advent of Large Language Models (LLMs) presents a new opportunity for rule learning. Given a few examples and task context, LLMs can directly generate rules in natural language(Ellis, 2023; Singh et al., 2022). Their linguistic richness and contextual alignment facilitate the articulation of fine-grained conditions, relations and exceptions, covering patterns that are difficult to capture with manually defined predicates. However, existing methods often focus on the iterative optimization of a single rule(Qiu et al., 2023) or the independent generation and simple aggregation of multiple rules(Yang et al., 2023; Zhou et al., 2024; Yang et al., 2024b), failing to jointly learn and compress the rules as a cohesive set. Concurrently, probabilistic approaches in the context of LLMs remains largely unexplored, as LLMs' ability to robustly learn the rule weights from data and apply them to perform reliable probabilistic reasoning during inference is not systematically examined yet. This gap motivates our pursuit of a framework that combines the generative and expressive capabilities of LLMs with the probabilistic composition and calibration of traditional methods to produce a flexible, robust and reusable rule set.

To achieve this synergy, we propose **RLIE**, a unified framework for LLM-based rule learning that integrates **R**ule generation, **L**ogistic regression, **I**terative refinement and **E**valuation. First, in the rule generation stage, an LLM produces a set of independent, individually assessable natural language rules based on a few examples and the task context. Second, for probabilistic combination, a regularized logistic regression model is employed to perform global weighting and selection on the rules, yielding a compact, interpretable rule set with importance weights. Third, in the iterative refinement stage, challenging examples are identified based on the prediction errors. These examples, along with the current rule set, are fed back to the LLM to prompt reflective generation of improved rules until convergence. Finally, a evaluation of two primary inference strategies is conducted: (1) direct inference using the weighted model, and (2) injecting the rule information into an LLM. For the latter, we systematically examine three levels of information: (i) providing only the rule, (ii) providing the rules with their weights, and (iii) providing the rules, weights and the linear model's prediction as a reference. The layered design allows for a systematic evaluation of quality of rules and analysis of the trade-offs in accuracy and robustness for different ways of using rules. Using our framework, we can get high quality rules in natural languages, with optimal ways of utilizing them to perform rule based reasoning. The main contribution of this paper are as follows:

1. We propose **RLIE**, a unified paradigm for LLM-based probabilistic rule learning. It combines the natural language rule generation of LLMs with the global weighting and selection of regularized logistic regression to produce a self-consistent, collaborative and auditable rule set.

2. We design a hierarchical set of inference strategies to systematically compare direct inference with the linear combiner against three methods of injecting information into an LLM: *rules-only, rules+weights* and *rules+weights+linear prediction reference*. We provide empirical analysis and practical guidance regrading their respective trade-offs in terms of accuracy, robustness and calibration.

3. On multiple real-world datasets, **RLIE** achieves superior over all performance compared to a range of LLM-based methods. The learnt rule sets are more compact and semantically clearer, prompting knowledge discovery and human-AI consensus.

## 2 RELATED WORK

### 2.1 CLASSICAL RULE LEARNING AND ITS APPLICATIONS

**Rule Learning**. Canonical approaches to rule learning include Inductive Logic Programming (ILP)(Cropper et al., 2022), which is centered on search and constraint optimization; association and pattern mining methods that rely on heuristics and pruning; and differentiable neuro-symbolic methods(Qiao et al., 2021; Glanois et al., 2022; Yang et al., 2024a) that couple symbolic rules with continuous representations. A common objective across these paradigms is to produce a set of human-readable and reusable discriminative rules from a structured candidate space. However, they are fundamentally dependent on a predefined predicate space.

**Rule Organization and Reasoning**. In the traditional paradigm, the interaction among rules is determined by their organization and the semantics of the inference process. One approach is the use of ordered rule lists or decision lists(Yang et al., 2017; Xu et al., 2024), which employ sequential matching to explicitly encode relationships of priority, exception, and default cases. This is well-suited for scenarios with clear sequential or override logic. Another approach involves unordered rule sets(Qiao et al., 2021; Yang & van Leeuwen, 2022), which support the parallel integration of multiple pieces of evidence, facilitating global trade-offs and model compression. Within the unordered setting, deterministic aggregation methods (e.g., logical OR/AND, thresholding, majority voting) are simple and intuitive but offer limited flexibility when faced with local conflicts, coverage gaps, or inter-rule dependencies. Consequently, many works have introduced probabilistic aggregation to model uncertainty and improve calibration. A common and interpretable implementation is the linear log-odds model, which treats the binary satisfaction of each rule $r_j(x) \in$

$\{0, 1\}$ on a sample $x$ as a feature and learns a weighted linear combination(Ruczinski et al., 2003):

$$\Pr(y = 1 \mid x) = \sigma(\beta_0 + \sum_j \beta_j r_j(x)).$$

Our work builds upon this foundation. Traditional methods typically assume rules are expressed in a fixed, executable symbolic form, allowing for direct evaluation of their satisfaction. In contrast, our approach focuses on rules expressed in natural language, where satisfaction must be determined by an LLM. We adopt a two-level design: at the "local judgment" level, we leverage the expressive power and adaptability of LLMs to interpret individual rules; at the "global aggregation" level, we retain the robust and calibratable probabilistic linear combiner from the classical paradigm. This hybrid design captures the flexible semantics of natural language rules while ensuring the overall inference process remains transparent and robust.

## 2.2 LLM-BASED RULE LEARNING AND ITS APPLICATIONS

**Rule Learning**. Large Language Models (LLMs) can directly propose hypotheses in natural language. A representative approach involves a "single-hypothesis refinement" loop, where the model maintains only the current best hypothesis and iteratively improves its quality and interpretability through feedback from validation sets or external executors(Qiu et al., 2023). In contrast, other methods adopt a "multi-hypothesis set" approach. For example, HypoGeniC(Zhou et al., 2024) generates multiple candidate hypotheses from data samples, maintains a top-$k$ set, and uses training signals to filter and update this set, ultimately forming an interpretable hypothesis bank. These two strategies exemplify the trade-off between deep optimization of a single hypothesis and the parallel retention of multiple diverse hypotheses.

**Rule Organization and Reasoning**. Rules expressed in natural language are typically reasoned over by the LLM that generated them. Existing work often passes one or more rules as context to an LLM, demonstrating that such information can enhance its reasoning capabilities(Zhang et al., 2024; Zhou et al., 2024). However, as the rules become more numerous or complex, the stability and instruction-following fidelity of LLMs can become unreliable. The black-box nature of LLMs makes it challenging to analyze the organization and interaction of these rules.

Our work is distinct from existing LLM-based rule learning approaches in that we are the first to explicitly combine LLMs with probabilistic methods to learn a set of weighted rules. This integration enables more principled sample selection during the iterative refinement stage and leads to more flexible and robust inference. To better analyze the synergy between this new rule paradigm and LLMs, we systematically compare direct inference using the probabilistic combiner with a hierarchy of strategies for injecting the rules, their weights, and the combiner's predictions into an LLM. This allows us to provide empirical analysis and practical guidance on the trade-offs in accuracy and robustness.

## 3 METHOD

We consider a binary classification task on a labeled dataset $\mathcal{S} = \{(x_i, y_i)\}_{i=1}^{N}$, where $x_i \in \mathcal{X}$ is a natural language text and $y_i \in \{0, 1\}$ is its corresponding label. The dataset $\mathcal{S}$ is partitioned into a training set $\mathcal{S}_{\text{tr}}$, a validation set $\mathcal{S}_{\text{val}}$ and a test set $\mathcal{S}_{\text{te}}$. To perform rule based reasoning, our objective is to learn a set of natural language rules $\mathcal{H}^{\star} = \{h_1, \ldots, h_m\}$ and a set of weights $\theta$ that aggregate the rules to model the discriminative relationship between $x$ and $y$. Here, $m \leq H$ where $H$ is a predefined capacity limit, and $h_i$ is a rule in natural language. Our method contains four stages, and the whole procedure is illustrated in Figure 1:

1. **R**ule Generation: We initialize an empty rule set $\mathcal{H} := \emptyset$. An LLM is prompted with a randomly selected subset of $\mathcal{S}_{\text{tr}}$ to generate candidate rules, which are then filtered based on coverage to form the initial rule set $\mathcal{H}^{(1)}$.

2. **L**ogistic Regression: To obtain the optimal weights to combinatorially use the generated rules, logistic regression is performed to learn the rule weights $\theta = (\boldsymbol{\beta}, b)$ on $\mathcal{S}_{\text{tr}}$ with hyperparameters tuned using $\mathcal{S}_{\text{val}}$.

3. **I**terative Refinement: Hard examples are selected from $\mathcal{S}_{\text{tr}}$ based on prediction errors. These samples, along with the current rule set $\mathcal{H}^{(t)}$, are fed back to the LLM to generate a set of new rules to revise existing rules. The union of newly generated rules and old rules can be directly used as $\mathcal{H}^{(t+1)}$, if it doesn't exceed the capacity limit $H$, otherwise we need to discard some rules based on their classification accuracy. The rule weights $\theta^{(t+1)}$ are also updated. This process is repeatedly executed for several times to iteratively refine the rules, while the whole process is monitored on $\mathcal{S}_{\text{val}}$ for early stopping.

Figure 1: The pipeline of the RLIE framework. **(1) Rule Generation:** An LLM proposes an initial set of rules from a small sample of training data. **(2) Logistic Regression:** A regularized logistic regression model is trained on the full training set to learn probabilistic weights for the current rules. **(3) Iterative Refinement:** Prediction errors from the regression model are used to mine hard examples, which are then used to prompt the LLM to refine or generate new rules. This cycle continues until performance on a validation set converges. **(4) Evaluation:** The learnt rules as well as weights are evaluated with different inference strategies. Detailed explanations of different strategies are in Section 3.4.

4. **E**valuation: After getting generated rules, we conduct a comprehensive evaluation on $\mathcal{S}_{\text{te}}$ using two classes of inference strategies: direct prediction using the linear combiner, and injecting the learnt rules and their weights into an LLM for augmented reasoning, to evaluate the quality of rules with different inference strategies.

## 3.1 RULE GENERATION

**Initialization and Candidate Generation**. In the first iteration, we initialize the rule set as empty, $\mathcal{H}^{(0)} := \emptyset$. We randomly sample $k$ examples from $\mathcal{S}_{\text{tr}}$ and prompt an LLM to generate $h$ candidate rules (where $h \leq H$), denoted as $\mathcal{H}^{(1)}_{\text{cand}} = \{h^{(1)}_1, \dots, h^{(1)}_h\}$.

**Individual Rule Application**. For any samples $x_i$ and candidate rule $h^{(1)}_j$, we prompt an LLM to obtain a local, ternary judgment

$$z^{(1)}_{i,j} = \text{LLM}\left(x_i, h^{(i)}_j\right) \in \{-1, 0, +1\},$$

as $z_{i,j}$ means the evaluation results of applying the $j$-th rule on the $i$-th sample where $+1$ indicates a positive prediction, $-1$ a negative prediction and $0$ signifies the rule is not applicable ("abstain"). Allowing abstention is crucial for explicitly modeling rule coverage, reducing misclassifications, and enabling sparse, robust combinations in the subsequent stage.

**Coverage-based Filtering and Rule Set Update**. For each candidate rule $h^{(1)}_j$, we compute its coverage on the training set as

$$\text{Cov}_{\text{tr}}\left(h^{(1)}_j\right) = \frac{1}{N_{\text{tr}}} \sum_{i=1}^{N_{\text{tr}}} \mathbb{I}\left[z^{(1)}_{i,j} \neq 0\right],$$

and discard rules with coverage lower than a predefined threshold $\gamma$. After $\mathrm{Cov}_{\mathrm{tr}}$ based filtering, the valid rules generated here is denoted as $\mathcal{H}_{\mathrm{new}}^{(1)}$, and the rule set is updated as $\mathcal{H}^{(1)} = \mathcal{H}^{(0)} \cup \mathcal{H}_{\mathrm{new}}^{(1)}$.

## 3.2 LOGISTIC REGRESSION

**Feature Construction**. For any iteration $t$, given a rule set $\mathcal{H}^{(t)} = \{h_1^{(t)}, \ldots, h_{m^{(t)}}^{(t)}\}$ where $m^{(t)} \leq H$, we define a mapping $\Phi^{(t)}$ from an input sample to a vector of rule judgments

$$\Phi^{(t)} : \ \mathcal{X} \to \{-1, 0, +1\}^{m^{(t)}}, \qquad \Phi^{(t)}(x_i) = \mathbf{z}_i^{(t)} = \left[z_{i,1}^{(t)}, \ldots, z_{i,m^{(t)}}^{(t)}\right]^\top.$$

**Weight Learning**. The probability of a sample $x_i$ being classified as positive is modeled by a logistic regression function, which is defined as below

$$p^{(t)}(x_i; \theta^{(t)}) = \sigma\big((\Phi^{(t)}(x_i))^\top \boldsymbol{\beta}^{(t)} + b^{(t)}\big), \qquad \sigma(u) = (1 + e^{-u})^{-1}$$

where $\sigma$ is the sigmoid function and $\theta^{(t)} = (\boldsymbol{\beta}^{(t)}, b^{(t)})$ are the model parameters. We learn these parameters by minimizing the cross-entropy loss on $\mathcal{S}_{\mathrm{tr}}$ with Elastic Net (Zou & Hastie, 2005) regularization:

$$\boldsymbol{\beta}^{(t)}, b^{(t)} = \arg\min_{\boldsymbol{\beta}^{(t)}, b^{(t)}} \frac{1}{|\mathcal{S}_{\mathrm{tr}}|} \sum_{x_i \in \mathcal{S}_{\mathrm{tr}}} \mathcal{L}(y_i, p^{(t)}(x_i; \theta^{(t)})) + \lambda\Big(\alpha\|\boldsymbol{\beta}^{(t)}\|_1 + \tfrac{1-\alpha}{2}\|\boldsymbol{\beta}^{(t)}\|_2^2\Big),$$

where $\mathcal{L}$ is the binary cross-entropy loss. The $L_1$ penalty encourages sparsity (rule selection), while the $L_2$ penalty enhances robustness. The regularization hyperparameters $(\lambda, \alpha)$ are selected via stratified K-fold cross-validation on $\mathcal{S}_{\mathrm{val}}$. The final parameters $\hat{\theta}^{(t)} = (\hat{\boldsymbol{\beta}}^{(t)}, \hat{b}^{(t)})$ are obtained by refitting the model on the entire $\mathcal{S}_{\mathrm{tr}}$ with the selected hyperparameters.

**Label Prediction**. After getting rule weights, the prediction for a sample $x_i$ can be calculated as

$$\hat{y}_i^{(t)} = \mathbb{I}\big[p^{(t)}(x_i; \hat{\theta}^{(t)}) \geq \tau\big] \in \{0, 1\}, \qquad \text{with } \tau = 0.5.$$

## 3.3 ITERATIVE REFINEMENT

In the first iteration, $\mathcal{H}^{(1)}$ and $\hat{\theta}^{(1)}$ are obtained on randomly selected samples. In the subsequent iterations, given $\mathcal{H}^{(t)}$ and $\hat{\theta}^{(t)}$ with $t \geq 1$, to generate $\mathcal{H}^{(t+1)}$ and $\hat{\theta}^{(t+1)}$, instead of relying on randomly selected samples, we can enhance the rule generation by targeting hard examples.

**1. Hard Example Selection**. For each example $x_i \in \mathcal{S}_{\mathrm{tr}}$, we compute its predicted probability $\hat{p}_i^{(t)} = p^{(t)}(x_i; \hat{\theta}^{(t)})$ and define its prediction error as $d_i = \|\hat{p}_i^{(t)} - y_i\|$. We identify the top-$k$ samples with the highest error $I_{\mathrm{hard}}^{(t)} = \mathrm{TopK}\left(\{(i, d_i)\}_{i=1}^{\|\mathcal{S}_{\mathrm{tr}}\|}, k\right)$.

**2. New Rule Generation**. The selected hard examples $I_{\mathrm{hard}}^{(t)}$ are presented to the LLM along with the current rule set $\mathcal{H}^{(t)}$. The LLM is prompted to reflect on the errors and either revise existing rules or generate new ones that cover novel perspectives. The $h$ newly generated rules are filtered for coverage, resulting in the set $\mathcal{H}_{\mathrm{new}}^{(t+1)}$.

**3. Rule Set Update**. The new rules are merged with the existing set: $\mathcal{H}_{\mathrm{tmp}}^{(t+1)} = \mathcal{H}^{(t)} \cup \mathcal{H}_{\mathrm{new}}^{(t+1)}$. If the size of this temporary set does not exceed the capacity $H$, it can be directly used as $\mathcal{H}^{(t+1)}$, otherwise we have to prune the rules by ranking them based on their individual accuracy on the validation set, retaining only the top $H$ rules to form $\mathcal{H}^{(t+1)}$.

**4. Parameter Update and Termination**. With the updated rule set $\mathcal{H}^{(t+1)}$, we calculate $\hat{\theta}^{(t+1)}$ in the same way as discussed in Section 3.2, after which we can monitor the overall performance on $\mathcal{S}_{\mathrm{val}}$. Rules will be refined iteratively, with each iteration trying to improve the performance on hard examples of previous version of rule sets. The process terminates if the overall performance on $\mathcal{S}_{\mathrm{val}}$ fails to improve by a margin $\delta$ for $p$ consecutive iterations or if the maximum number of iterations $R_{\max}$ is reached. The final model $(\mathcal{H}^\star, \hat{\theta}^\star)$ corresponds to the checkpoint with the best validation performance.

## 3.4 EVALUATION

We systematically compare four inference strategies on the test set $\mathcal{S}_{\mathrm{te}}$, categorized into two main approaches: direct inference using the linear combiner and LLM-augmented inference with varying levels of information.

**(E1) Linear-only**. The final prediction is made directly by the logistic regression model, with $\hat{y}_i^{\text{final}} = \mathbb{I}\big[p(x_i; \hat{\theta}^\star) \geq 0.5\big]$, where $p(\cdot; \hat{\theta}^\star)$ is the learnt probabilistic combiner. This serves as a classic baseline for weighted rule sets.

**(E2) LLM + Rules**. The LLM is provided with only the texts of the rules in $\mathcal{H}^\star$ and the input $x_i$, and is tasked with making a prediction where $\hat{y}_i^{\text{final}} = \text{LLM}(x_i, \mathcal{H}^\star)$. This evaluates the LLM's intrinsic ability to aggregate rule-based evidence, mirroring common practices in existing work.

**(E3) LLM + Rules + Weights**. In addition to the rule texts, the LLM is also provided with the learnt weights $\hat{\theta}^\star$ and the final prediction becomes $\hat{y}_i^{\text{final}} = \text{LLM}(x_i, \mathcal{H}^\star, \hat{\theta}^\star)$. This tests whether the LLM can leverage explicit probabilistic signals to guide its reasoning, reflecting a "probabilistic internalization" of rule importance.

**(E4) LLM + Rules + Weights + Linear Prediction**. The LLM receives all information from generated rules and corresponding weights, plus the prediction $\hat{y}_i$ from the linear-only model as a reference, so the prediction becomes $\hat{y}_i^{\text{final}} = \text{LLM}(x_i, \mathcal{H}^\star, \hat{\theta}^\star, \hat{y}_i)$. This strategy assesses whether a potentially imperfect but calibrated rule-based system can provide a valuable external signal to enhance the LLM's reasoning in complex scenarios where linear assumption may not fully hold.

## 4 EXPERIMENT SETUP

### 4.1 DATASET

To evaluate our method, we select six real-world tasks from the HypoBench(Liu et al., 2025)Language benchmark, including Deception Detection (**Review**), Mental Stress Detection (**Dreddit**), News Headline Engagement (**Headlines**), Paper Citations (**Citations**), AI-generated Content Detection (**LLM Detect**) and Retweets (**Retweets**). Each of the task can be viewed as a binary classification task, with text information as inputs and binary classification results as output (e.g. given a review, we need to evaluate whether it is deceptive or not). Details of each dataset are in Appendix A.3.

### 4.2 BASELINE

We implemented and compared RLIE against several representative and competitive methods within a unified framework to comprehensively evaluate its performance and robustness. (1) **Zero-shot Inference**: The LLM directly predicts the output for a given sample based only on the task description, without access to any examples or rules. (2) **Zero-shot Generation**: The LLM generates a set of candidate rules based solely on the task description. The single best-performing rule, as determined on the validation set, is then used for inference on the test set. (3) **IO Refinement** (Qiu et al., 2023): This method follows a "propose-select-refine" loop. In each round, the LLM generates a batch of candidate hypotheses, the best one is selected based on its performance on a validation or error set, and it is then used as a seed for the next round of refinement. This process is repeated for a fixed number of iterations. (4) **HypoGeniC** (Zhou et al., 2024): This approach maintains and dynamically expands a library of hypotheses. For each sample, the top-$k$ hypotheses are selected to make a prediction. Mismatched samples are added to an error set, which triggers the generation of new hypotheses to improve coverage. The hypothesis library is continuously updated based on a reward signal.

### 4.3 EXPERIMENTAL DETAILS

In our experiments, each dataset was partitioned into fixed-size training, validation, and test sets of 200, 200, and 300 samples, respectively. All methods used the same data splits and a fixed random seed to ensure reproducibility. We report *Accuracy* and *Macro-F1* score. Each experiment was repeated at least three times, and we report the mean and standard deviation of the results. All experiments involving LLMs utilized `gpt-4o-mini` with the temperature set to $1 \times 10^{-5}$ to ensure deterministic outputs. Prompt details can be found in Appendix E. As for hyperparameters, the capacity of the rule set was set to $H = 10$. In each iteration, we sampled $k = 20$ hard examples and generated $h = 5$ new rules. The minimum coverage threshold for rules was set to $\gamma = 0.2$. The implementations of IO Refinement and HypoGeniC followed the default hyperparameter settings of the original papers. Further details are provided in Appendix A.

Table 1: Overall performance comparison (Accuracy / Macro-F1, $\times 10^{-2}$) on full datasets. RLIE (Ours) uses the Linear-Only inference strategy. The best results among generalizable methods are **bolded**. Note that LoRA achieves high scores on simple tasks but fails to generalize on complex reasoning tasks.

| Method | Backbone | Reviews | Dreaddit | Headline | Citations | LLM Detect | Retweets |
|---|---|---|---|---|---|---|---|
| Zero-shot | DeepSeek-V3 | 53.9 / 42.3 | 64.7 / 60.0 | 61.2 / 59.6 | 62.5 / 50.0 | 82.4 / 82.0 | 63.5 / 62.6 |
| Few-shot (ICL) | DeepSeek-V3 | 65.3 / 65.1 | 63.8 / 58.5 | 62.0 / 61.8 | 60.4 / 49.8 | 80.4 / 79.6 | 57.7 / 53.9 |
| Zero-shot Gen | DeepSeek-V3 | 65.4 / 64.8 | 67.2 / 63.9 | 57.5 / 57.4 | 46.1 / 47.7 | 63.1 / 57.6 | 58.1 / 58.1 |
| IO Refinement | DeepSeek-V3 | 65.9 / 65.4 | 78.5 / 78.1 | 62.0 / 61.1 | 54.2 / 51.0 | 83.6 / 83.4 | 57.1 / 56.1 |
| HypoGeniC | DeepSeek-V3 | 69.1 / 69.3 | 80.5 / 80.5 | 59.9 / 60.1 | 46.9 / 49.3 | 85.2 / 85.1 | 61.9 / 61.8 |
| LoRA Finetune | Qwen3-8B | *94.1 / 94.1* | 54.4 / 54.4 | 51.5 / 51.5 | 52.1 / 51.0 | *99.7 / 99.8* | 51.4 / 51.4 |
| **RLIE (Ours)** | Qwen3-Next-80B | 68.3 / 67.8 | 81.1 / 81.1 | 61.1 / 60.9 | 56.3 / 55.0 | 87.6 / 87.6 | 61.9 / 61.8 |
| **RLIE (Ours)** | Qwen3-235B | **71.5 / 71.4** | 79.9 / 79.9 | 60.6 / 60.4 | 61.5 / 60.6 | 88.3 / 88.3 | **66.5 / 66.5** |
| **RLIE (Ours)** | DeepSeek-V3 | 70.9 / 70.7 | **82.3 / 82.3** | **67.0 / 67.0** | **64.6 / 63.0** | **90.7 / 90.7** | 65.7 / 65.6 |

## 5 RESULTS

This section presents empirical evaluation of our framework. First, we compare the overall performance of RLIE against competitive baselines. Second, we conduct an ablation study on different inference strategies to identify the most effective way to utilize the learned probabilistic rules.

### 5.1 MAIN RESULTS

The empirical results, summarized in Table 1, demonstrate the effectiveness and robustness of RLIE compared to several strong baselines across six real-world datasets. For this comparison, RLIE's performance is evaluated using its direct inference strategy (Linear-only), where the learned logistic regression model acts as the classifier. While some baseline methods may attain the top performance on one or two individual datasets, RLIE consistently ranks within the top two in terms of both Accuracy and F1 Score. This demonstrates that the RLIE framework can be generalized to diverse data distributions and is effective at capturing underlying data patterns. As shown in the results table, some methods exhibit high variance and instability despite their strong performance on certain tasks (e.g., IO Refinement). In contrast, our method achieves high performance while maintaining stability, underscoring its robustness.

A horizontal comparison of the different methods reveals several interesting insights. It is noteworthy that Zero-shot Inference outperforms many of the more complex baselines in several scenarios. This may be attributed to the extensive knowledge acquired by the LLM during pre-training, which is sufficient for solving certain tasks. However, this internal inference process is inherently unexplainable, and it is difficult to rule out the influence of potential data leakage from the pre-training corpus. This observation highlights both the challenge and the necessity of using rule-based learning to achieve interpretable reasoning. HypoGeniC learns a set of rules independently and fails to account for the interactions between them, resulting in suboptimal performance. By iteratively refining a single rule, IO Refinement achieves respectable results, which validates the value of providing annotated examples and employing an iterative optimization process. It is worth mentioning that in some cases, IO Refinement outperforms RLIE. This could be because the strategy of generating only a single rule forces it to be more generalizable, allowing it to better handle unseen scenarios. However, this single-rule approach also limits the method's expressiveness, leading to poor performance in cases that require a more complex rule set. Our RLIE framework strikes a balance: by continuously learning from hard examples, it enhances the generalizability of the learned rule set, while the strategy of maintaining multiple rules significantly boosts its expressiveness. This allows our method to achieve excellent and robust performance with low variance across diverse datasets. A detailed case study is illustrated in Appendix B.

### 5.2 HOW TO BEST UTILIZE RULES?

To determine the optimal strategy for leveraging the learned probabilistic rules, we evaluated the four distinct inference methods detailed in Section 3.4. One might hypothesize that an LLM, with its vast pre-trained knowledge, could synthesize the rule set, their learned weights, and even a reference prediction to produce superior judgments. However, the empirical results, presented in Table 2, are counterintuitive.

The most striking finding is that the simplest strategy, Linear-only, where the logistic regression model makes the final prediction, achieves the best performance on nearly all datasets. It outperforms all three LLM-based inference

Table 2: Impact of Inference Strategies (F1 Score, $\times 10^{-2}$). **(E1) Linear-Only** consistently outperforms strategies that inject rules back into the LLM (E4).

| Backbone | Strategy | Rev. | Dread. | Head. | Cit. | Det. | Ret. |
|---|---|---|---|---|---|---|---|
| DeepSeek V3.2 | (E1) Linear | **70.7** | 82.3 | **67.0** | **63.0** | **90.7** | **65.6** |
| | (E2) Rules | 63.1 | 76.6 | 66.8 | 56.2 | 89.6 | 64.3 |
| | (E3) +Wgt | 64.0 | 77.1 | 65.0 | 53.5 | 85.0 | 61.8 |
| | (E4) +Full | 68.6 | **82.4** | 66.8 | 55.9 | 89.3 | 64.6 |
| Qwen3 235B | (E1) Linear | **71.4** | **79.9** | **60.4** | **60.6** | 88.3 | 66.5 |
| | (E2) Rules | 64.2 | 76.7 | 59.7 | 60.3 | 88.4 | 63.2 |
| | (E3) +Wgt | 63.2 | 77.1 | 58.2 | 60.1 | 87.1 | 65.0 |
| | (E4) +Full | 66.6 | 79.5 | 59.8 | 54.0 | 87.9 | 65.7 |

strategies in most cases, particularly in terms of F1 score where it is the top performer across all six tasks. This highlights the high quality of the rule set curated by RLIE and the effectiveness of the probabilistic combiner.

Injecting more information into the LLM does not yield consistent benefits. Providing the LLM with rule weights (LLM + Rules + Weights) over just the rule texts does not lead to stable performance gains, suggesting that the LLM struggles to effectively internalize and apply probabilistic information to refine its reasoning. Most surprisingly, providing the LLM with the linear model's own (often correct) prediction as a reference (LLM + Rules + Weights + Linear Prediction) frequently results in performance degradation compared to simply using the linear model's prediction directly. This suggests that the LLM, when tasked with integrating qualitative probabilities and multiple form evidences into reasoning, can be led astray, potentially overwriting a correct judgment with an incorrect one. This finding aligns with existing research indicating that LLMs can be inconsistent when required to strictly adhere to complex, explicit instructions, showing the deficiency of LLMs to perform fine controlled inference.

# 6 DISCUSSION

Our hierarchical evaluation reveals a central finding: achieving stable, globally consistent reasoning by prompting an LLM with a set of explicitly weighted rules is non-trivial. The simplest "Linear-only" strategy, which uses the learned logistic regression model as the final classifier, performed best on nearly all datasets and achieved the highest F1 score across the board. In contrast, injecting the rule texts, their weights, and even the linear model's reference prediction back into the LLM often failed to yield stable improvements and, in some cases, degraded performance. This result aligns with the observation that LLMs excel at semantic generation and interpretation but are less reliable at fine-grained, controlled probabilistic integration. This suggests that a more robust paradigm involves a clear division of labor: leveraging LLMs for local, semantic tasks such as judging individual rules and generating new candidates, while entrusting a transparent and calibratable probabilistic combiner—in our work, a logistic regression model with elastic net regularization—with the global tasks of weighting and selection.

From a broader perspective, using natural language rules as the fundamental unit of reasoning bypasses the rigid dependency on predefined predicate spaces found in traditional methods, offering greater expressiveness for open-ended semantics and unstructured data. At the same time, this approach avoids overwhelming the LLM's context with complex probabilistic details from an entire rule set, thereby preserving its ability to make stable judgments at the single-rule level. Unlike frameworks that model interactions at the predicate level (e.g., Markov Logic Networks), we advocate for enhancing the capabilities of the global combiner while keeping the local interface—the LLM's ternary judgments and coverage information—fixed. This linear log-odds combiner can be naturally extended to more sophisticated models, such as (i) interpretable additive models (e.g., GAMs) to capture rule interactions; (ii) message passing on a factor graph to globally harmonize rule confidences; (iii) Bayesian logistic regression with hierarchical priors (e.g., ARD, Laplace) for joint uncertainty and sparsity modeling; or (iv) techniques like Platt scaling or isotonic regression to explicitly calibrate the LLM's "abstain" outputs. These extensions are fully compatible with the current RLIE architecture. Overall, we propose a replicable engineering principle: LLMs are responsible for semantic generation and local judgment, while traditional probabilistic learning methods handle global aggregation and uncertainty management. On this foundation, stronger graphical or additive structures and more advanced inference algorithms can be progressively introduced to build more flexible and powerful, yet still interpretable and calibratable, neuro-symbolic systems.

## 7 CONCLUSION

In this paper, we addressed the critical challenge of learning robust and interpretable rule sets from data. While Large Language Models (LLMs) have created new possibilities for generating rules in natural language, existing methods have struggled with the principled aggregation and calibration of these rules. We introduced **RLIE**, a unified framework that synergizes the rule-generation capabilities of LLMs with the global optimization of probabilistic models. By integrating rule generation, regularized logistic regression, and error-driven iterative refinement, RLIE produces a compact and calibrated set of weighted, natural language rules. Our experiments demonstrate that this hybrid approach is highly effective. However, directly using LLMs to apply weighted rules degrades the performance of downstream tasks, which further unveils the limitations of applying LLMs in rule based reasoning. Our work underscores the value of a neuro-symbolic division of labor, where LLMs handle local semantic tasks and classical models manage global probabilistic reasoning. The paper investigates the potentials and limitations of using LLMs for inductive reasoning tasks, proposing a unified framework which integrates LLMs with classic probabilistic rule combination methods, paving the way for more building reliable AI and contribute to knowledge discovery.

## 8 ETHIC STATEMENT

This research adheres to the ICLR Code of Ethics. The datasets used in our experiments are derived from the HypoBench benchmark, which is composed of publicly available data from prior academic research. To the best of our knowledge, personally identifiable information has been anonymized in these datasets. We acknowledge that the rules generated by our method reflect the patterns and potential societal biases present in the underlying training data. While a primary goal of our work is to enhance interpretability, this also means that any learned biases will be transparent. We caution that the application of such rule-based systems, particularly in sensitive domains like mental health analysis or persuasion, should be accompanied by careful human oversight to mitigate the risk of misuse or the perpetuation of harmful stereotypes. Our work is intended to contribute to the development of more transparent and auditable AI systems.

## 9 REPRODUCIBILITY STATEMENT

We are committed to ensuring the reproducibility of our research. The complete source code for the RLIE framework, along with the scripts for running all experiments, will be made publicly available upon publication. All datasets used in this study are from the public HypoBench benchmark, and we will provide the exact data splits to facilitate direct comparison. A detailed description of our experimental setup, including model specifications, baseline implementations, and primary hyperparameters, is provided in Section 4. A comprehensive list of all prompts used to interact with the LLM for rule generation, judgment, and inference is included in the Appendix E. The code will be released once the paper is accepted.

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

## A  MORE DETAILS

In this section, we provide additional information regarding the baseline methods used for comparison and the specific validation procedures employed in our experiments.

### A.1  BASELINES

To contextualize the performance of **RLIE**, we compare it against two state-of-the-art LLM-based rule learning frameworks:

- **IO Refinement**: Proposed by Qiu et al. (2023), this method follows a "propose-select-refine" loop. It begins by generating candidate rules and iteratively refines a single best hypothesis based on feedback from its performance on a set of examples.
- **HypoGeniC**: Introduced by Zhou et al. (2024), this framework maintains a dynamic library of hypotheses. It uses a reward mechanism inspired by multi-armed bandits to balance the exploration of new rules and the exploitation of well-performing ones, updating its rule set based on performance on training instances.

### A.2  VALIDATION

Our experimental setup follows a standardized procedure, but certain details regarding the use of the validation set for baselines and the handling of smaller datasets warrant clarification.

For datasets where the total number of available samples was less than the specified 200 for training and 200 for validation, we utilized all available data for the respective splits.

The baseline methods handle the validation set differently according to their original designs. Since the algorithms for **IO Refinement** and **Zero-shot Generation** do not inherently use a validation set during their rule generation phase, we implemented a final selection step for a fair comparison: after these methods completed their final iteration, we selected the single rule that achieved the highest performance on our designated validation set to be used for testing. In contrast, the **HypoGeniC** algorithm does not involve a validation set in its update loop; its mechanism for updating and pruning the hypothesis library relies on rewards calculated from performance on the training batches. Therefore, for HypoGeniC, the validation set was not used during its training and refinement process.

### A.3  DATASET

**Deception Detection (Reviews)**. The objective is to distinguish between genuine and deceptive hotel reviews. Deceptive reviews, often written by paid individuals, may appear plausible but typically lack specific details. The primary challenge lies in identifying subtle linguistic cues, such as exaggerated sentiment or vague descriptions, which are difficult for even human evaluators to discern reliably.

**Persuasive Argument Prediction (Dreddit)**. The objective is to detect mental stress signals from Reddit posts across different communities. These posts, sourced from social media, often contain complex personal narratives that reflect an individual's psychological state. The primary challenge lies in investigating specific linguistic features indicative of mental stress, identifying subtle patterns in the text that effectively signal the presence of distress.

**News Headline Engagement (Headlines)**. Given two headlines for the same news event, the model must predict which one is more likely to attract clicks. The task examines how linguistic choices in news writing—such as novelty, specificity, or emotional framing—influence reader behavior. The often minimal difference between headlines makes this task highly challenging.

**Paper Citations (Citations)**. This task involves predicting whether an academic paper will achieve a high citation count based on its title and abstract. It focuses on features related to academic impact, such as the generality of the research problem, the novelty of the contributions, or the timeliness of the research topic. Unlike social media tasks, this task emphasizes long-term value signals in academic writing.

**AI-generated Content Detection (LLM Detect)**. This task requires determining whether a given text was written by a human or generated by an LLM. The dataset contains human- and machine-written texts based on the same prompts, evaluating a model's ability to capture stylistic, structural, or semantic differences indicative of AI-generated content and formulate them as interpretable rules.

**Retweets (Retweets)**. The input is a pair of tweets, and the task is to predict which one will be retweeted more. Influential factors include a tweet's conciseness, emotional intensity, and the mention of specific individuals or orga-

nizations. Due to the highly stochastic nature of social media propagation, this task places stringent demands on the explanatory power and stability of the learned rules.

## B    CASE STUDY

In this section, we provide a qualitative analysis of the iterative refinement process on the **Retweets** task. Table 3 illustrates the evolution of the rule set across three rounds. In each iteration, RLIE generates new candidate rules (marked in red) to address hard examples, while the logistic regression component assigns weights to retain effective rules (marked in blue) and prune ineffective or redundant ones (marked in gray). As shown, the semantic quality of the rules improves from generic observations to specific, high-confidence patterns, corresponding to a steady improvement in training accuracy (from 0.625 to 0.700).

Table 3: Iterative Rule Refinement Process (Retweet Task)

| Rd | Prompt (Sample) | Rule Bank (Selected Rules) | Metric |
|---|---|---|---|
| 0 | **T1:** WATCH LIVE: #House debates issue of #health insurance policies eliminated because of #Obamacare http://t.co/gNkJkVS3IR 

 **T2:** WATCH LIVE: #House voting on bill to restore #health insurance policies canceled under #ObamaCare http://t.co/gNkJkVS3IR 

 *Label: 1* | 1. Tweets that use stronger emotional language or dramatic framing (e.g., "grossly negligent," "messy fast") are more likely to be retweeted. (**W:0.035**) 

 2. Tweets that include vivid details, quotes, or specific examples outperform those that offer general summaries. (**W:0.008**) 

 3. Tweets that highlight novelty, urgency, or time-sensitive action (e.g., "RT to win!") are more likely to be shared. (**W:0.000**) 

 *Bias: 0.005* | Acc: 0.625 
 F1: 0.625 |
| 1 | **T1:** Andrew C. McCarthy: The Scheme behind the Obamacare Fraud... 

 **T2:** Like all swindles, Obamacare cannot work if its targeted victims figure out the endgame... 

 *Label: 1* | 1. Tweets that prioritize clarity and immediate comprehension over stylistic flair or irony are favored in news/public affairs... (**W:0.045**) 

 2. Tweets that use stronger emotional language or dramatic framing are more likely to be retweeted... (**W:0.034**) 

 3. Tweets that highlight novelty, urgency, or time-sensitive action are more likely to be shared... (**W:0.000**) 

 *Bias: 0.001* | Acc: 0.680 
 F1: 0.679 |
| 2 | **T1:** NEW VID!!    :D *****MEET MY DAD!***** http://t.co/MyUFfCZKsP RT ? :) 

 **T2:** what do you think about the SNEAK PEEK of the CHRISTMAS SONG?!    :) *****MEET MY DAD!***** ... 

 *Label: 1* | 1. Tweets that use personal voice or self-referential framing ("my view," "we're live") are more retweetable when they convey authenticity... (**W:0.223**) 

 2. Tweets that prioritize clarity and immediate comprehension over stylistic flair or irony are favored... (**W:0.178**) 

 3. Tweets with conversational, informal, or playful tone are more retweetable than those using formal language... (**W:0.000**) 

 *Bias: 0.104* | Acc: 0.700 
 F1: 0.699 |

## C    PARAMETER STUDY

Table 4: Sensitivity analysis of the coverage threshold $\gamma$ on the Headline dataset (Backbone: DeepSeek-V3). Performance remains robust for $\gamma \in [0.1, 0.5]$.

| Coverage Threshold ($\gamma$) | Accuracy | F1 Score |
|---|---|---|
| 0.1 | 66.7 | 66.7 |
| 0.2 | 67.0 | 67.0 |
| 0.3 | **67.3** | **67.2** |
| 0.4 | 66.5 | 66.5 |
| 0.5 | 66.5 | 66.5 |
| 0.6 | 65.5 | 65.5 |
| 0.7 | 66.5 | 66.5 |
| 0.8 | 65.6 | 65.6 |
| 0.9 | 64.3 | 64.2 |

## D    LLM USAGE

Large Language Models (LLMs) were used as an assistive tool in preparing this manuscript. The core intellectual contributions, including the research ideas, experimental design, and analysis, are entirely the work of the human

authors. The LLM's role was limited to language polishing and coding assistance. The authors have reviewed and edited all LLM-generated content and take full responsibility for the final manuscript.

## E  PROMPTS

Here we are providing the prompts we are using to conduct all the experiments. For the clarity of illustration, here we are only showing prompts for the Retweet task, while all the prompts can be access after this paper is accepted.

```
multi_content: |
    The first tweet: ${first_tweet}
    The second tweet: ${second_tweet}
    Final answer: The ${label} tweet got more retweets.
```

Figure 2: Prompt for providing observations.

```
system: |-
    You are a social media expert. You are an expert at determining which tweet
        will be retweeted more.
    Given a set of observations, you want to generation hypotheses that will help
        predict which tweet out of a pair of tweets is more likely to be retweeted.
    Please note that the paired tweets are about the same content and are posted by
         the same user, so you should focus on the wording difference between the
        two tweets in each pair.
    Please propose ${num_hypotheses} possible hypotheses.
    Please generate them in the format of:
    1. [hypothesis]
    2. [hypothesis]
    ...
    ${num_hypotheses}. [hypothesis].
    Please make the hypotheses general enough to be applicable to new observations.
user: |-
    We made some observations:
    ${observations}
    Generate hypotheses that are useful for predicting which tweet out of a pair of
        tweets is more likely to be retweeted.
    Please note that the paired tweets are about the same content and are posted by
         the same user, so you should focus on the wording difference between the
        two tweets in each pair.
    Please propose ${num_hypotheses} possible hypotheses.
    Please generate them in the format of:
    1. [hypothesis]
    2. [hypothesis]
    ...
    ${num_hypotheses}. [hypothesis].
    Proposed hypotheses:
```

Figure 3: Prompt for the first iteration of rule generation.

```
system: |-
  You are a social media expert focused on maximizing retweet engagement.
  Given misclassified tweet pairs and prior hypotheses, your goal is to rethink and
      propose ${num_hypotheses} new, more accurate hypotheses about which tweet in
      a pair will earn more retweets.
  Please note that the paired tweets share the same content and author, so
      concentrate on wording differences, framing, and presentation.
  Generate the hypotheses in the format of:
  1. [hypothesis]
  2. [hypothesis]
  ...
  ${num_hypotheses}. [hypothesis].
  Please make the hypotheses general enough to be applicable to new observations.
user: |-
  We have tweet pairs that previous hypotheses predicted incorrectly:
  ${observations}

  Here are some of the prior hypotheses for reference:
  ${hypotheses_text}

  Please generate new hypotheses that better capture which tweet in each pair will
      get more retweets.
  You may refine the previous hypotheses (like tightening conditions, adding
      exceptions, or rephrasing), or introduce new hypotheses to cover new,
      distinct angles when prior ones are insufficient or misaligned.

  Propose ${num_hypotheses} possible hypotheses.
  Generate them in the format of 1. [hypothesis], 2. [hypothesis], ... ${
      num_hypotheses}. [hypothesis].
  Proposed hypotheses:
```

Figure 4: Prompt for iterative refinement.

```
system: |-
  You are a social media expert.
  Given a pair of tweets, you are asked to predict which tweet will be retweeted
      more.
  Please note that the paired tweets are about the same content and are posted by
      the same user, so you should focus on the wording difference between the two
      tweets.
  From past experiences, you learned a pattern.
  Now, at each time, you should apply a learned pattern to a pair of tweets and
      determine which one will get more retweets.
  Give an answer. Respond with exactly one of: first, second, or not applicable.
  Give your final answer in the format of {Final answer: first} or {Final answer:
      second}.
user: |-
  Pattern: ${hypothesis}
  The first tweet: ${first_tweet}
  The second tweet: ${second_tweet}

  Given the pattern you learned above, predict which one of the two tweets will get
       more retweets.
  Think step by step.
  First step: Consider if the pattern can be applied to analyze the textual
      difference between the two tweets.
  Third step: Based on the pattern, which tweet is more likely to get more retweets
      ? If it does not apply, say so explicitly.
  Final step: Give your final answer in the format of {Final answer: first}, {Final
       answer: second} or {Final answer: not applicable}.
  Final answer:
```

Figure 5: Prompt for single hypothesis inference.

```
system: |-
  You are a social media expert.
  Given a pair of tweets, you are asked to determine which will get more retweets.
  From past experiences, you learned some patterns.
  You need to determine whether each of the patterns holds for the current pair of
      tweets, and also predict which tweet will get more retweets.
  Give your final answer in the format of {Final answer: first} or {Final answer:
      second}.
user: |-
  Our learned patterns: ${hypotheses}
  The first tweet: ${first_tweet}
  The second tweet: ${second_tweet}

  Given the patterns you learned above, predict which one will get more retweets.
  Think step by step.
  First step: Think about which patterns can be applied to these tweets.
  Second step: Based on the applicable patterns, which tweet is likely to get more
      retweets?
  Give your final answer in the format of {Final answer: first} or {Final answer:
      second}.
```

Figure 6: Prompt for "LLM + Rules" style inference.

```
system: |-
  You are a social media expert.
  Given a pair of tweets, you are asked to determine which will get more retweets.
  We trained a linear regression model on the training set to obtain a collection
      of weighted patterns plus a bias term. The learned weight reflects how
      strongly the pattern contributes to predicting "first" vs "second".
  Review the weighted hypotheses, consider how the bias interacts with them, and
      use the regression model's suggested label only as a reference.
  Give your final answer in the format of {Final answer: first} or {Final answer:
      second}.
user: |-
  Our learned weighted patterns (the weight's magnitude reflects the pattern's
      importance):
  ${weighted_hypotheses}

  Bias term (a constant offset added regardless of pattern activations; a positive
      bias means the model is overall more inclined toward ${pos_label}, while a
      negative bias leans toward ${neg_label}):
  ${bias}

  The first tweet: ${first_tweet}
  The second tweet: ${second_tweet}

  Given the patterns you learned above, predict which one will get more retweets.
  Think step by step.
  First step: Think about which patterns can be applied to these tweets.
  Second step: Decide which tweet is likely to get more retweets, you can used the
      weighted patterns and bias as reference.
  Final step: give your final answer in the format of {Final answer: first} or {
      Final answer: second}.
```

Figure 7: Prompt for "LLM + Rules + Weights" style inference.

```
system: |-
  You are a social media expert.
  Given a pair of tweets, you are asked to determine which will get more retweets.
  We trained a linear regression model on the training set to obtain a collection
      of weighted patterns plus a bias term. The learned weight reflects how
      strongly the pattern contributes to predicting "first" vs "second". The
      regression model also outputs a referenced label for the review, which should
       be treated as a suggestion.
  Review the weighted hypotheses, consider how the bias interacts with them, and
      use the regression model's suggested label only as a reference.
  Give your final answer in the format of {Final answer: first} or {Final answer:
      second}.
user: |-
  Our learned weighted patterns (the weight's magnitude reflects the pattern's
      importance):
  ${weighted_hypotheses}

  Bias term (a constant offset added regardless of pattern activations; a positive
      bias means the model is overall more inclined toward ${pos_label}, while a
      negative bias leans toward ${neg_label}):
  ${bias}

  The first tweet: ${first_tweet}
  The second tweet: ${second_tweet}

  We have used the regression model to get a referenced label: ${model_prediction}

  Given the patterns you learned above, predict which one will get more retweets.
  Think step by step.
  First step: Think about which patterns can be applied to these tweets.
  Second step: Decide which tweet is likely to get more retweets, you can used the
      weighted patterns, bias, predicted label as reference.
  Final step: give your final answer in the format of {Final answer: first} or {
      Final answer: second}.
```

Figure 8: Prompt for "LLM + Rules + Weights + Linear Prediction" style inference.