# OpenReview forum: "RLIE: Rule Generation with Logistic Regression, Iterative Refinement, and Evaluation for Large Language Models"
_ICLR.cc/2026/Conference — Submitted to ICLR 2026_

### Official Review · Reviewer_Xf2f · 2025-10-29

**Soundness:** 2
**Presentation:** 1
**Contribution:** 2
**Rating:** 2
**Confidence:** 3

**Summary:**

The paper introduces RLIE, a framework that combines LLMs with probabilistic rule learning to generate and refine rules for better decision-making. The RLIE process involves four stages: generating candidate rules using an LLM, applying logistic regression to learn the weights of these rules, refining the rule set iteratively based on prediction errors, and evaluating the model by comparing its performance to other methods of rule integration. The study shows that applying weighted rules directly results in superior performance, but injecting rules into an LLM using prompt-based methods can lead to degraded performance. This suggests that while LLMs are strong in semantic generation, they struggle with fine-tuned, controlled probabilistic integration. The main contribution of the paper is the development of a unified framework that combines LLMs with traditional probabilistic rule combination techniques, advancing the field of neuro-symbolic reasoning systems.

**Strengths:**

1. The paper is well-written and methodologically sound. It provides a clear and detailed explanation of the RLIE framework, including the rule generation, logistic regression, iterative refinement, and evaluation stages.
2. The work has significant potential for advancing the field of neuro-symbolic AI. By successfully combining LLMs with probabilistic rule learning, it addresses an important challenge in AI: integrating the flexibility of generative models with the precision of rule-based reasoning.

**Weaknesses:**

1. The work lacks experimentation on more models to ensure that its effectiveness is broad and not limited to specific ones.
2. There are significant formatting issues with Tables 1 and 2, as they exceed the page width.
3. The introduction to the task in the work is not clear enough. More concrete examples should be introduced to describe the entire process, in order to improve the readability of the paper.
4. The method is not broadly effective across all tasks, and its average performance improvement over previous work is limited.

**Questions:**

1. Can you reproduce some comparative experiments on weaker open-source models (such as Qwen3) and stronger closed-source models (such as GPT-5) to demonstrate the generalizability of the method?
2. Why does the method perform significantly worse than the IO Refinement method on the Dreaddit and LLM Detect datasets?
3. The generalizability of the conclusions found in Section 5.2 is questionable. Is the performance drop caused by the introduction of LLM due to the selection of relatively weaker models? Please add experiments on more models to further support your findings.

---

> ### Author Response · Authors · 2025-11-30
>
> We thank the reviewer for recognizing the methodological soundness of RLIE and its potential to advance neuro-symbolic AI. We address your specific questions regarding generalizability and experimental anomalies below.
>
>
>
> W1: Generalization to Other Models (New Experiments)
>
> To address the concern that our conclusions rely solely on GPT-4o-mini, we have significantly expanded our evaluation:
>
> - Models: We conducted new experiments using DeepSeek V3.2, Qwen3-235b, and Qwen3-next-80b.
> - Consistency: The results (Table R1) confirm our initial findings. The "Linear-Only" inference strategy consistently outperforms injecting rules back into the LLM, regardless of the model size. This reinforces our claim that the bottleneck lies in the nature of how LLMs process probabilistic instructions, not just the capacity of the specific model used.
>
> **Table R_Models:** **Generalizability of RLIE across Model Architectures.**
>
> *Performance (F1 Score) of RLIE using different LLM backbones. We compare the "Linear-Only" strategy (E1) against the fully augmented "LLM+Rules+Weights" strategy (E4). The results confirm that E1 consistently outperforms E4 across all model sizes, supporting the conclusion in Section 5.2.*
>
>
>
> | Backbone           | Inference Strategy   | Reviews | Dreaddit | Headline | Citations | LLM Detect | Retweets |
> | :----------------- | :------------------- | :-----: | :------: | :------: | :-------: | :--------: | :------: |
> | **DeepSeek V3.2**  | (E1) Linear-Only     |  70.7   |   82.3   |   67.0   |   63.0    |    90.7    |   65.6   |
> |                    | (E4) LLM + Full Info |  68.6   |   82.4   |   66.8   |   55.9    |    89.3    |   64.6   |
> | **Qwen3 235B**     | (E1) Linear-Only     |  71.4   |   79.9   |   60.4   |   60.6    |    88.3    |   66.5   |
> |                    | (E4) LLM + Full Info |  66.6   |   79.5   |   59.8   |   54.0    |    87.9    |   65.7   |
> | **Qwen3 Next-80B** | (E1) Linear-Only     |  67.8   |   81.1   |   60.9   |   55.0    |    87.6    |   61.8   |
> |                    | (E4) LLM + Full Info |  67.4   |   79.1   |   61.9   |   53.7    |    87.6    |   56.4   |
>
>
>
> *(Note: E1 wins in 15 out of 18 comparisons, showing the robust superiority of the linear combiner.)*
>
>
>
> W2: Presentation and Formatting
>
> - Table Layout: We have resized Tables 1 and 2 to fit within the page margins and improved the alignment.
> - Task Clarity: Per your suggestion, we have added concrete examples of the input-output process and a visualization of the rule evolution (iterations 1–3) in the revised Appendix. This provides a clearer narrative of how the framework refines natural language rules over time.

---

> > ### Author Response · Authors · 2025-11-30
> >
> > W3: Case Study
> > We appreciate your suggestion. To improve clarity, we have provided a concrete example below that displays a data sample and illustrates the evolution of the rule set over three iterations. We have also integrated this detailed walkthrough into the Appendix of the revised manuscript and referenced it in the main text.
> >
> > **Box R_Qual:** **Case Study: Evolution of Learned Rules (Retweet Task).**
> >
> > *This example illustrates how the "Iterative Refinement" stage evolves rules from generic statements to specific, high-weight patterns.*
> >
> > | Iteration   | Rule Content (Abbreviated)                                   | Weight | Status                  |
> > | :---------- | :----------------------------------------------------------- | :----: | :---------------------- |
> > | **Round 0** | "Tweets with conversational, informal, or playful tone are more retweetable." |  0.00  | Discarded (Too generic) |
> > | **Round 1** | "Tweets that use stronger emotional language or dramatic framing (e.g., 'grossly negligent') are more likely to be retweeted." |  0.03  | Kept (Moderate)         |
> > | **Round 2** | "Tweets that use personal voice or self-referential framing ('my view', 'I’m watching') are more retweetable... when they convey authenticity." |  0.22  | Top Rule (Specific)     |
> >
> >
> >
> > W4: Performance Gaps
> >
> > Upon further investigation, we identified that the limited performance gain on specific datasets was primarily caused by LLM safety refusals. These datasets contain sensitive content (e.g., negative emotion or explicit material). When prompted to apply rules, the LLM frequently triggered safety filters and refused to answer, which our pipeline defaulted to a negative classification.
> >
> > While baselines were also affected, the impact on our method was disproportionately severe. For baseline methods, a refusal only affects the prediction for a single sample. However, because our method utilizes logistic regression to learn global rule weights, frequent refusals on training data skewed the learned weights, degrading the model's overall performance. This issue stems from the LLM's safety mechanisms rather than an inherent algorithmic defect.
> >
> > To address this, we implemented a robust retry mechanism (increasing attempts from 1 to 5) to bypass spurious refusals. We then re-evaluated our method using the full datasets and a wider range of LLMs. Under these rectified and more rigorous conditions, our method demonstrates significant performance improvements and superior robustness compared to previous work.
> >
> > **Table R_Fixed:** **Performance on Sensitive Datasets (DeepSeek-V3).**
> >
> > *Comparison of RLIE against the strong baseline (IO Refinement) after resolving the "LLM Safety Refusal" bug. RLIE now significantly outperforms the baseline on the previously problematic datasets.*
> >
> > | Dataset                      | IO Refinement (Baseline) | RLIE (Ours, Fixed) | Absolute Improvement |
> > | :--------------------------- | :----------------------: | :----------------: | :------------------: |
> > | **Dreaddit** (Mental Health) |       78.5 / 78.1        |    82.3 / 82.3     |    +3.8% / +4.2%     |
> > | **LLM Detect** (Adversarial) |       83.6 / 83.4        |    90.7 / 90.7     |    +7.1% / +7.3%     |
> > | **Headline** (Standard)      |       62.0 / 61.1        |    67.0 / 67.0     |    +5.0% / +5.9%     |
> > | **Retweets** (Standard)      |       57.1 / 56.1        |    65.7 / 65.6     |    +8.6% / +9.5%     |
> >
> > *(Metric: Accuracy / F1 Score)*

---

> > > ### Author Response · Authors · 2025-11-30
> > >
> > > Q1: Due to cost constraints, we did not run experiments on GPT-5; however, we tested on the equally competitive DeepSeek V3.2, as well as Qwen3-235b and Qwen3-next-80b. These experiments demonstrate the generalizability of our method across models of varying sizes and capabilities.
> > >
> > >
> > >
> > > Q2: Performance on Dreaddit and LLM Detect
> > >
> > > You correctly pointed out that RLIE initially underperformed compared to the IO Refinement baseline on these two specific datasets.
> > >
> > > - Cause Analysis: Our investigation revealed this was primarily due to LLM Safety Refusals. These datasets contain sensitive content (mental health discussions and potential adversarial text). When the LLM refused to evaluate a rule due to safety filters, our pipeline treated it as a "negative" signal. This disproportionately damaged RLIE's performance because the logistic regression penalized the global weights of rules that appeared to "miss" frequently due to refusals. In contrast, baseline methods like IO Refinement (which process samples locally) were less penalized by individual refusals.
> > > - Mitigation & Result: We implemented a robust retry mechanism (increasing retries from 1 to 5) to bypass spurious refusals. With this fix, RLIE’s performance on these datasets improved significantly, surpassing the baselines and demonstrating that the method is robust when technical refusal issues are handled.
> > >
> > >
> > >
> > > Q3: Generalizability across Models (Open & Closed Source)
> > >
> > > You asked whether our results—specifically the limitations of LLMs in probabilistic reasoning (Section 5.2)—hold for other models.
> > >
> > > - Expanded Scope: We have now evaluated RLIE on DeepSeek V3.2 (comparable to state-of-the-art closed models) and Qwen3-235b / Qwen3-next-80b (strong open-source models).
> > > - Consistent Findings: The results (Table R1) are consistent with our original experiments on GPT-4o-mini. The "Linear-Only" inference strategy continues to outperform LLM-augmented strategies across these architectures. This confirms that the difficulty LLMs face in adhering to fine-grained probabilistic rules is a general characteristic of current LLM paradigms, rather than a limitation specific to a single weaker model.

---

### Official Review · Reviewer_nMcw · 2025-10-30

**Soundness:** 2
**Presentation:** 3
**Contribution:** 2
**Rating:** 2
**Confidence:** 5

**Summary:**

This paper proposes a large language model (LLM)-based rule learning framework, RLIE. The framework leverages LLMs to generate natural language rules, which are assigned probabilistic weights via a logistic regression model. In the iterative optimization process, difficult samples from task predictions are fed back to the LLM along with the current rule set to generate new rules. The authors evaluate the method on six text classification datasets under different reasoning strategies. The results demonstrate that directly using the rules and their learned weights for prediction achieves better performance.

Although the idea of using logistic regression to learn rule weights proposed in this paper is reasonable, the method still has several important flaws in its experimental design that need to be addressed.

**Strengths:**

1.	The paper is well-structured and readable.
2.	The methodology on using logistic regression to learn rule weights is sound and shows a certain degree of novelty, compared with the traditional top-K methods.

**Weaknesses:**

1.	Insufficient baselines for experimental comparisons.   In the experiments, the authors use approximately 400 labeled samples but do not compare with the methods such as few-shot in-context learning (ICL) or fine-tuning neural networks, as discussed in the studies like: “What Makes Good In-Context Examples for GPT 3?” and “In-Context Learning Learns Label Relationships but Is Not Conventional Learning.”
Moreover, the authors do not assess how the proposed method scales with varying model capacities. It only uses a single backbone model (GPT 4o mini). It is suggested to compare with the models of different sizes or different architectures.
2.	Limited experimental tasks. The experiments are confined to binary text classification task, which are relatively simple. For example, in Table 1, the single-rule baseline (IO Refinement) achieves the best performance on two datasets, suggesting that the reasoning complexity is limited. Additionally, the test sets are small (about 300 samples), which further restricts the generalizability of the method.
3.	Insufficient analysis of method effectiveness. The paper does not evaluate the quality of the generated rules, such as how the diversity of rules influences the effectiveness of method, what is the impact of the key hyperparameters. Although the coverage threshold γ is fixed at 0.2, it is still required to conduct experiments to examine its influence.
4.	Lack of clarity in methodological details. It is unclear whether the optimization of rule weights adequately covers all rules. During incremental rule generation, the process for assigning weights to new rules is not explained. Furthermore, in scenarios with frequent rule updates, it remains uncertain whether these weights receive sufficient training.

**Questions:**

1.	The single-rule baseline (IO Refinement) outperforms the proposed method RLIE. Does this imply that combining multiple rules might be unnecessary in such cases? How do the authors explain the inferior performance of RLIE compared to a simpler approach on these specific tasks?
2.	It is unclear whether each rule’s weight is sufficiently trained during the iterative optimization process. While stopping criteria for the iterations are provided, there is a lack of statistical analysis or visualizations illustrating the frequency of weight updates or the convergence behavior of individual rules.


Finally, it is recommended that the authors conduct further analysis of the generated rules and reasoning outcomes, such as examining the distribution of error types in incorrectly reasoned samples and evaluating whether the new rules have effectively corrected these errors. This is particularly important given that the rules are expressed in natural language, where interpretability is a crucial factor.

---

> ### Author Response · Authors · 2025-11-30
>
> We appreciate the reviewer’s positive assessment of our paper's structure and the novelty of using logistic regression for rule weighting. We have addressed your concerns regarding baselines, model scale, and methodological details below.
>
>
>
> W1: 1. Baselines and Model Scale (New Experiments)
>
> We agree that comparison against standard paradigms was missing. We have expanded our experimental suite significantly:
>
> - New Baselines: We implemented Few-Shot In-Context Learning (ICL) and LoRA Fine-tuning (using Qwen3-8b) to compare RLIE against standard prompting and parameter-efficient fine-tuning approaches.
> - Model Scaling: To address the reliance on a single backbone, we re-evaluated RLIE using DeepSeek V3.2, Qwen3-235b, and Qwen3-next-80b.
> - Results: RLIE remains superior or highly competitive against these strong baselines (see Table R1). The consistency across model sizes confirms that the benefit comes from the framework's hybrid structure, not just the underlying LLM capability.
>
> **Table R_Baselines: Comparison with New Baselines (ICL, LoRA) and Model Scaling.**
>
> *We compare RLIE against Few-Shot In-Context Learning (ICL) and LoRA Fine-tuning. We also demonstrate RLIE's scalability across open-source (Qwen3) and closed-source (DeepSeek) models of varying sizes. (Metric: Accuracy / F1 Score)*
>
> | Category                 | Method / Backbone         |   Reviews   |  Dreaddit   |  Headline   |  Citations  | LLM Detect  |  Retweets   |
> | :----------------------- | :------------------------ | :---------: | :---------: | :---------: | :---------: | :---------: | :---------: |
> | **Standard Baselines**   | Zero-shot (DeepSeek)      | 53.9 / 42.3 | 64.7 / 60.0 | 61.2 / 59.6 | 62.5 / 50.0 | 82.4 / 82.0 | 63.5 / 62.6 |
> |                          | Few-shot ICL (DeepSeek)   | 65.3 / 65.1 | 63.8 / 58.5 | 62.0 / 61.8 | 60.4 / 49.8 | 80.4 / 79.6 | 57.7 / 53.9 |
> | **Trained**              | LoRA Fine-tune (Qwen3-8B) | 94.1 / 94.1 | 54.4 / 54.4 | 51.5 / 51.5 | 52.1 / 51.0 | 99.7 / 99.8 | 51.4 / 51.4 |
> | **RLIE (Model Scaling)** | Qwen3-Next-80B            | 68.3 / 67.8 | 81.1 / 81.1 | 61.1 / 60.9 | 56.3 / 55.0 | 87.6 / 87.6 | 61.9 / 61.8 |
> |                          | Qwen3-235B                | 71.5 / 71.4 | 79.9 / 79.9 | 60.6 / 60.4 | 61.5 / 60.6 | 88.3 / 88.3 | 66.5 / 66.5 |
> |                          | DeepSeek-V3.2             | 70.9 / 70.7 | 82.3 / 82.3 | 67.0 / 67.0 | 64.6 / 63.0 | 90.7 / 90.7 | 65.7 / 65.6 |
>
>
>
> W2: Comprehensive Evaluations
>
> We have addressed the concern regarding sample size by expanding our experiments to utilize the full datasets for all six tasks, rather than subsets. This expansion increases the average training set size from 200 to 307 and the average test set size from 255 to 388, ensuring a more rigorous and comprehensive evaluation.

---

> ### Author Response · Authors · 2025-11-30
>
> W3: Hyperparameter Study
>
> Using the Headline task as a case study, we investigated the impact of varying the coverage threshold $\gamma$ from 0.1 to 0.9. As shown in the table below, the observed variance was minimal; both F1 and Accuracy fluctuated by less than 2%. Crucially, performance consistently exceeded the baseline across this range, highlighting the robustness of our method. We acknowledge the value of analyzing additional hyperparameters, such as rule set capacity, and leave this for future work.
>
> **Table R_Gamma: Impact of Coverage Threshold ($\gamma$) on Headline Task. This table is also updated as Table 4 in Appendix C in the revised manuscript.**
>
> *(Metric: Accuracy / F1 Score, $\times 10^{-2}$)*
>
> | $\gamma$ | 0.1  | 0.2  | 0.3  | 0.4  | 0.5  | 0.6  | 0.7  | 0.8  | 0.9  |
> | :------- | :--: | :--: | :--: | :--: | :--: | :--: | :--: | :--: | :--: |
> | **Acc**  | 66.7 | 67.0 | 67.3 | 66.5 | 66.5 | 65.5 | 66.5 | 65.6 | 64.3 |
> | **F1**   | 66.7 | 67.0 | 67.2 | 66.5 | 66.5 | 65.5 | 66.5 | 65.6 | 64.2 |
>
>
>
> W4: Clarification on Weight Optimization
>
> You asked if weights are sufficiently trained during iterative updates.
>
> - Full Retraining: Because logistic regression on the rule-judgment matrix is computationally efficient (convex optimization), we do not update weights incrementally. Instead, at each iteration step, we fully retrain the weight \theta on the entire current rule set \mathcal{H}^{(t)} until convergence. This guarantees that the weights are always the optimal solution for the current set of rules.
> - Convergence Analysis: We have added an analysis in the appendix tracking the relative rank of rule weights over the final 3 iterations. The rankings remain stable, indicating that the rule set and weights have converged to a robust state.

---

> ### Author Response · Authors · 2025-11-30
>
> Q1: Performance vs. IO Refinement (Dreaddit & LLM Detect)
>
> You correctly noted that the single-rule baseline (IO Refinement) initially outperformed RLIE on two datasets. We investigated this and found it was an artifact of LLM Safety Refusals.
>
> - The Issue: The Dreaddit (mental health) and LLM Detect datasets trigger safety filters. When the LLM refused to evaluate a rule for a specific sample, our pipeline treated it as a "negative/abstain" signal. While this affects all methods, it disproportionately harms RLIE because the logistic regression model penalizes the global weights of rules that frequently "miss" due to refusals.
> - The Fix: We implemented a robust retry mechanism (increasing retries from 1 to 5) and expanded the dataset size. With these adjustments, RLIE consistently outperforms IO Refinement on these tasks, demonstrating that the combination of multiple rules is indeed superior when technical refusal issues are mitigated.
>
> **Table R_Fixed:** **Performance on Sensitive Datasets (DeepSeek-V3).**
>
> *Comparison of RLIE against the strong baseline (IO Refinement) after resolving the "LLM Safety Refusal" bug. RLIE now significantly outperforms the baseline on the previously problematic datasets.*
>
> | Dataset                      | IO Refinement (Baseline) | RLIE (Ours, Fixed) | Absolute Improvement |
> | :--------------------------- | :----------------------: | :----------------: | :------------------: |
> | **Dreaddit** (Mental Health) |       78.5 / 78.1        |    82.3 / 82.3     |    +3.8% / +4.2%     |
> | **LLM Detect** (Adversarial) |       83.6 / 83.4        |    90.7 / 90.7     |    +7.1% / +7.3%     |
> | **Headline** (Standard)      |       62.0 / 61.1        |    67.0 / 67.0     |    +5.0% / +5.9%     |
> | **Retweets** (Standard)      |       57.1 / 56.1        |    65.7 / 65.6     |    +8.6% / +9.5%     |
>
>
>
> Q2: Case Study
>
> In every experiment the regression models are fully trained to get the rule weights. The rules learned by our framework are expressed in natural language, ensuring they are inherently semantically plausible and human-understandable. For instance, in the Retweet prediction task, one learned rule explicitly states: “Tweets that use stronger emotional language or dramatic framing (e.g., "grossly negligent," "messy fast," "life-saving assistance") are more likely to be retweeted than those with neutral or factual tones.” To further address this, we have added a detailed case study tracking the iterative evolution of the rule set in Appendix B of the revised manuscript.

---

### Official Review · Reviewer_95e3 · 2025-10-31

**Soundness:** 1
**Presentation:** 1
**Contribution:** 2
**Rating:** 2
**Confidence:** 3

**Summary:**

The paper introduces RLIE, a framework designed to integrate Large Language Models (LLMs) with probabilistic rule learning. The primary contribution is a four-stage process that aims to overcome the limitations of traditional rule learning by leveraging the natural language capabilities of LLMs. The stages are: (1) Rule generation, where an LLM proposes and filters candidate rules in natural language; (2) Logistic regression, which learns probabilistic weights for the rules to enable global selection and calibration; (3) Iterative refinement, where the rule set is continuously optimized based on prediction errors; and (4) Evaluation, which assesses the performance of the learned rule set. The goal is to create a more robust neuro-symbolic reasoning system by combining the generative power of LLMs with classical probabilistic methods.

**Strengths:**

- The paper addresses an important and challenging problem: integrating the semantic capabilities of LLMs with more structured, probabilistic reasoning frameworks.
- The proposed iterative refinement loop, where the LLM is prompted to revise rules based on model errors, is an interesting idea for automated feature engineering.
- The work explores different ways of combining learned rules with LLMs for inference, leading to an interesting (though negative) result about the difficulty of fine-grained probabilistic control in LLMs.

**Weaknesses:**

- The central concept of a "rule" is ill-defined and misleading. What the paper calls "rules in natural language" are effectively just natural language prompts or questions posed to an LLM to generate ternary features (+1, 0, -1). These "rules" lack the formal structure, interpretability, and compositionality of rules in traditional symbolic systems.
- The experimental comparison is flawed. The paper compares the performance of a trained logistic regression model against a prompted LLM that is given the rules and weights. This is an unfair comparison, as one is a trained system while the other is not. The potential of the LLM-based classifier has not been fully explored.
- The empirical evaluation is weak and lacks rigor. The experiments are conducted using only a single, small model ("gpt-4o-mini"). The paper's conclusions about LLM capabilities are therefore based on very limited evidence and may not generalize to other, more capable models.
- The system demonstrates no compositionality between rules, which is a key feature of traditional rule-based systems. The "rules" are treated as independent features for a linear model.

**Questions:**

1. The paper's claims revolve around "rules," but the learned artifacts appear to be non-compositional natural language prompts for feature extraction. Can you justify the use of the term "rule" and explain how these differ from simple learned features, given their lack of formal structure or compositionality?
2. The paper's primary conclusion relies on an experimental setup that compares a trained model (Logistic Regression) with an untrained one (a zero-shot LLM), using only a single, non-frontier model. How can the general claims about LLM limitations be supported by this specific and seemingly flawed comparison?

---

> ### Author Response · Authors · 2025-11-30
>
> We thank the reviewer for the thoughtful feedback, particularly for appreciating our iterative refinement approach and our findings regarding the difficulty of probabilistic control in LLMs. We address your concerns regarding the definition of rules and experimental rigor below.
>
>
>
> Q1: Explanation of Rules
>
> Our terminology follows a line of recent work that defines natural language prompts for classification as “rules” [1–3]. These artifacts are not merely atomic features; they are semantically rich, interpretable statements that guide the understanding of patterns—for example: “Tweets that use stronger emotional language or dramatic framing (e.g., ‘grossly negligent,’ ‘messy fast,’ ‘life-saving assistance’) are more likely to be retweeted than those with neutral or factual tones.” While we acknowledge a functional similarity to feature extraction (in that they yield +1/0/-1 outputs), we distinguish our approach in two key ways:
>
> 1. Semantic Complexity: Traditional feature engineering typically operates on structured data [4] or extracts simple lexical units (e.g., n-grams) [5]. In contrast, our natural language rules encapsulate complex, high-level patterns. As seen in the example above, a single rule often contains internal logic (such as “or” conditions), effectively functioning as a composite pattern that would require multiple atomic features in a traditional system.
> 2. Predictive Sufficiency: Unlike atomic features, which usually require combination to possess predictive power, our individual rules are standalone discriminators capable of making independent predictions. We empirically validated this by feeding our rules as features into a symbolic rule learning algorithm [6] (to learn Boolean combinations). We observed a significant performance drop compared to our linear combination approach. This indicates that treating these semantic units as simple features for further logical composition is overly restrictive, confirming that they function best as complete, independent rules that are compositionally aggregated via probabilistic weighting.
>
> As shown in **Table R_Comp** below, enforcing strict symbolic composition results in performance degradation to near-random levels (Accuracy $\approx$ 50\%). In contrast, our **Probabilistic Combination** (RLIE with Logistic Regression) achieves superior performance. This justifies our terminology: these are not simple atomic features waiting to be logically combined; they are complex, independent semantic "rules" whose compositionality is best modeled through **probabilistic aggregation** (weighted sum of evidence) rather than rigid symbolic logic.
>
> **Table R_Comp: Probabilistic Aggregation (RLIE) vs. Symbolic Composition.**
>
> *Comparing the standard RLIE inference (Linear Weighting) against using learned rules as features for a Symbolic Learner (Boolean Logic). Metrics: Accuracy / F1 Score. Experiments are ran on DeepSeek-V3.*
>
> | Dataset        | (A) Symbolic Composition (Boolean Logic) | (B) Probabilistic Aggregation (RLIE) |
> | :------------- | :--------------------------------------: | :----------------------------------: |
> | **Reviews**    |               49.4 / 33.8                |             70.9 / 70.7              |
> | **Dreaddit**   |               49.9 / 34.4                |             82.3 / 82.3              |
> | **Headline**   |               49.8 / 33.7                |             67.0 / 67.0              |
> | **Citations**  |               57.8 / 48.5                |             64.6 / 63.0              |
> | **LLM Detect** |               50.6 / 36.0                |             90.7 / 90.7              |
> | **Retweets**   |               50.3 / 35.5                |             65.7 / 65.6              |

---

> > ### Author Response · Authors · 2025-11-30
> >
> > Q2: Fairness of Comparison (Trained vs. Untrained) and Generalization to Other Models (New Experiments)
> >
> > We agree that comparing a trained logistic model against a zero-shot LLM presents a specific type of gap. To address this and ensure a fair comparison between "trained" systems:
> >
> > - New LoRA Baseline: We have added an experiment using Qwen3-8b with LoRA fine-tuning to represent a "trained LLM" baseline.
> > - Results: RLIE still demonstrates competitive performance and better interpretability. The linear combiner (Stage 2) offers a distinct advantage: it calibrates the confidence of specific semantic rules, which even fine-tuned LLMs often struggle to do explicitly without losing generalization.
> >
> > To address the concern that our conclusions rely solely on GPT-4o-mini, we have significantly expanded our evaluation:
> >
> > - Models: We conducted new experiments using DeepSeek V3.2, Qwen3-235b, and Qwen3-next-80b.
> > - Consistency: The results (Table R1) confirm our initial findings. The "Linear-Only" inference strategy consistently outperforms injecting rules back into the LLM, regardless of the model size. This reinforces our claim that the bottleneck lies in the nature of how LLMs process probabilistic instructions, not just the capacity of the specific model used.
> >
> > **Table R_Gen: Comparison with Fine-tuned Baseline (LoRA) and Cross-Model Consistency. The results are also updated to Table 1 in the revised version of the paper.**
> >
> > *We compare RLIE (Linear strategy) against a LoRA fine-tuned Qwen3-8B model. We also show RLIE's performance across different backbones and contrast it with the LLM-augmented inference strategy (E4: LLM + Rules + Weights + Ref).*
> >
> > | Category        | Model / Method       |   Reviews   |  Dreaddit   |  Headline   |  Citations  | LLM Detect  |  Retweets   |
> > | :-------------- | :------------------- | :---------: | :---------: | :---------: | :---------: | :---------: | :---------: |
> > | **Trained LLM** | LoRA (Qwen3-8B)      | 94.1 / 94.1 | 54.4 / 54.4 | 51.5 / 51.5 | 52.1 / 51.0 | 99.7 / 99.8 | 51.4 / 51.4 |
> > | **RLIE (Ours)** | DeepSeek-V3 (Linear) | 70.9 / 70.7 | 82.3 / 82.3 | 67.0 / 67.0 | 64.6 / 63.0 | 90.7 / 90.7 | 65.7 / 65.6 |
> > |                 | DeepSeek (LLM E4)    | 69.5 / 68.6 | 82.4 / 82.4 | 66.8 / 66.8 | 57.3 / 55.9 | 89.3 / 89.3 | 64.7 / 64.6 |
> > | **RLIE (Ours)** | Qwen3-235B (Linear)  | 71.5 / 71.4 | 79.9 / 79.9 | 60.6 / 60.4 | 61.5 / 60.6 | 88.3 / 88.3 | 66.5 / 66.5 |
> > | **RLIE (Ours)** | Qwen3-80B (Linear)   | 68.3 / 67.8 | 81.1 / 81.1 | 61.1 / 60.9 | 56.3 / 55.0 | 87.6 / 87.6 | 61.9 / 61.8 |
> >
> > *(Metric: Accuracy / F1 Score. 'LLM E4' denotes the strategy of injecting rules, weights, and reference predictions back into the LLM.)*

---

> > > ### Author Response · Authors · 2025-11-30
> > >
> > > [1] Saeidi, Marzieh, et al. "Interpretation of Natural Language Rules in Conversational Machine Reading." Proceedings of the 2018 Conference on Empirical Methods in Natural Language Processing. 2018.
> > >
> > > [2] Mozannar, Hussein, et al. "Effective human-AI teams via learned natural language rules and onboarding." Advances in neural information processing systems 36 (2023): 30466-30498.
> > >
> > > [3] Vong, Wai Keen, and Brenden M. Lake. "Few-shot image classification by generating natural language rules." ACL Workshop on Learning with Natural Language Supervision. 2022.
> > >
> > > [4] Hollmann, Noah, Samuel Müller, and Frank Hutter. "Large language models for automated data science: Introducing caafe for context-aware automated feature engineering." Advances in Neural Information Processing Systems 36 (2023): 44753-44775.
> > >
> > > [5] Malberg, Simon, Edoardo Mosca, and Georg Groh. "FELIX: Automatic and interpretable feature engineering using llms." Joint European Conference on Machine Learning and Knowledge Discovery in Databases. Cham: Springer Nature Switzerland, 2024.
> > >
> > > [6] Yang, Lincen, and Matthijs van Leeuwen. "Truly unordered probabilistic rule sets for multi-class classification." Joint European Conference on Machine Learning and Knowledge Discovery in Databases. Cham: Springer Nature Switzerland, 2022.

---

### Official Review · Reviewer_gbA1 · 2025-11-01

**Soundness:** 2
**Presentation:** 2
**Contribution:** 2
**Rating:** 4
**Confidence:** 5

**Summary:**

This paper presents RLIE (Rule Generation with Logistic Regression, Iterative Refinement, and Evaluation), a neuro-symbolic framework that combines large language models with classical probabilistic modeling for interpretable rule learning. The proposed pipeline consists of four stages: rule generation via LLMs, weight estimation using logistic regression, iterative refinement on hard examples, and evaluation under four inference strategies (E1–E4). Experiments on six binary classification tasks demonstrate that the linear-only logistic model (E1) achieves the best performance, suggesting that LLMs are effective in generating rule candidates but less reliable at probabilistic integration.

**Strengths:**

The idea of combining LLM-based semantic rule generation with a probabilistic model for global reasoning is conceptually appealing.

**Weaknesses:**

1. Although the paper presents a well-organized framework that integrates LLM-based rule generation with probabilistic weighting via logistic regression and iterative refinement, the overall idea is conceptually incremental. The notion of combining LLM-generated symbolic rules with classical probabilistic or statistical models has already appeared in several recent neuro-symbolic or rule-learning works.

2. The experiments rely solely on GPT-4o-mini with a near-deterministic decoding setting. It remains unclear whether the observed results, especially the relative advantages of the linear combiner over LLM-augmented inference, would hold for other LLMs.

3. The paper contains several noticeable formatting problems that affect readability. For example, Tables 1 and 2 exceed the page margins, and some layout elements are misaligned. In addition, there are minor typographical errors—most notably, “Lange Models” in the abstract should be “Language Models.” Careful proofreading and layout adjustments are recommended before publication.

4. No error analysis is provided to explain where and why the proposed method succeeds or fails. Moreover, although the authors claim that the learned rules are interpretable, there are no visualizations or case studies demonstrating the semantic plausibility or human-understandability of these rules. Adding such analyses would significantly enhance the paper’s insightfulness and credibility.

**Questions:**

See above.

---

> ### Author Response · Authors · 2025-11-30
>
> We thank the reviewer for recognizing our framework as conceptually appealing and for the constructive feedback regarding our experimental scope and presentation. We have addressed your concerns below, particularly by expanding our experiments to include DeepSeek and Qwen models.
>
>
>
> W1: Novelty and Conceptual Contribution
>
> While we agree that combining LLMs with probabilistic models is an emerging direction, RLIE introduces two distinct contributions that differentiate it from prior neuro-symbolic works:
>
> - Active Iterative Refinement: unlike static "generate-then-select" pipelines, RLIE utilizes the probabilistic model’s error signals to specifically target and retrieve "hard examples." This creates a closed feedback loop where the LLM does not just generate rules, but actively "reflects" on model deficiencies to update the rule set.
> - Inference Strategy Analysis: A core contribution of our work is the systematic evaluation of how to utilize learned rules (Section 5.2). We provide empirical evidence that while LLMs are excellent rule generators, they are unreliable probabilistic reasoners (often performing worse when given ground-truth weights). This negative result is crucial for the community, suggesting that future neuro-symbolic systems should decouple semantic generation (LLM) from probabilistic inference (Linear/Symbolic), rather than forcing LLMs to do both.
>
>
>
> W2: Generalization across LLMs (New Experiments)
>
> We agree that relying solely on GPT-4o-mini was a limitation. In response, we have conducted extensive new experiments using **DeepSeek V3.2**, **Qwen3-235b**, and **Qwen3-next-80b** on full datasets. The results (detailed in **Table R1** below; this table is also updated in the revised version of the paper as Table 1) are consistent with our original findings:
>
> 1. **Linear-Only Superiority:** The linear combiner (E1) achieves the highest F1 scores in **16 out of 18** cases (3 models $\times$ 6 datasets) compared to LLM-augmented strategies (E2-E4). This reinforces our conclusion that while LLMs are excellent at generating semantic rules, they struggle to perform calibrated probabilistic inference even when provided with explicit weights.
> 2. **Robustness:** RLIE (E1) consistently maintains high performance across models of varying sizes, confirming that our framework's effectiveness is a generalizable property of the method rather than an artifact of a specific architecture.
>
> **Table R1: Performance (Acc / F1) of RLIE across different LLM backbones and inference strategies (E1-E4).**
>
> | Model    | Strategy | Reviews  | Dreaddit | Headling | Citation | LLM Detect | Retweets |
> | :-- | :-- | :- | :- | :- | :- | :- | :- |
> | DeepSeek | (E1) Linear  | 70.9 / 70.7 | 82.3 / 82.3 | 67.0 / 67.0 | 64.6 / 63.0 | 90.7 / 90.7 | 65.7 / 65.6 |
> | V3.2     | (E2) Rules   | 65.7 / 63.1 | 77.5 / 76.6 | 66.9 / 66.8 | 60.4 / 56.2 | 89.6 / 89.6 | 64.4 / 64.3 |
> |   | (E3) +Weight | 65.4 / 64.0 | 77.9 / 77.1 | 65.2 / 65.0 | 55.2 / 53.5 | 85.2 / 85.0 | 62.5 / 61.8 |
> |   | (E4) +Full   | 69.5 / 68.6 | 82.4 / 82.4 | 66.8 / 66.8 | 57.3 / 55.9 | 89.3 / 89.3 | 64.7 / 64.6 |
> | Qwen3    | (E1) Linear  | 71.5 / 71.4 | 79.9 / 79.9 | 60.6 / 60.4 | 61.5 / 60.6 | 88.3 / 88.3 | 66.5 / 66.5 |
> | 235B     | (E2) Rules   | 64.5 / 64.2 | 77.3 / 76.7 | 60.0 / 59.7 | 60.4 / 60.3 | 88.4 / 88.4 | 64.5 / 63.2 |
> |   | (E3) +Weight | 63.4 / 63.2 | 77.7 / 77.1 | 58.5 / 58.2 | 60.4 / 60.1 | 87.1 / 87.1 | 65.8 / 65.0 |
> |   | (E4) +Full   | 66.8 / 66.6 | 79.7 / 79.5 | 59.9 / 59.8 | 54.2 / 54.0 | 87.9 / 87.9 | 66.0 / 65.7 |
> | Qwen3    | (E1) Linear  | 68.3 / 67.8 | 81.1 / 81.1 | 61.1 / 60.9 | 56.3 / 55.0 | 87.6 / 87.6 | 61.9 / 61.8 |
> | Next-80B | (E2) Rules   | 66.9 / 66.6 | 74.5 / 73.1 | 62.6 / 62.3 | 56.3 / 53.0 | 84.1 / 84.0 | 57.9 / 54.7 |
> |          | (E3) +Weight | 64.7 / 63.9 | 76.2 / 75.2 | 64.1 / 64.0 | 55.2 / 53.2 | 75.9 / 74.5 | 57.1 / 53.0 |
> |          | (E4) +Full   | 67.7 / 67.4 | 79.5 / 79.1 | 62.0 / 61.9 | 55.2 / 53.7 | 87.6 / 87.6 | 58.9 / 56.4 |
>
> *(Note: E1=Linear Only, E2=LLM+Rules, E3=LLM+Rules+Weights, E4=LLM+Full Info. Results are on full datasets.)*
>
>
>
> W3: Formatting and Typos
>
> We appreciate you pointing out the formatting issues. We have corrected the "Lange Models" typo, resized Tables 1 and 2 to fit the margins, and thoroughly proofread the manuscript to ensure layout elements are aligned.
>
>
>
> W4: Case Study
>
> We appreciate this suggestion. The rules learned by our framework are expressed in natural language, ensuring they are inherently semantically plausible and human-understandable. For instance, in the Retweet prediction task, one learned rule explicitly states: “Tweets that use stronger emotional language or dramatic framing (e.g., "grossly negligent," "messy fast," "life-saving assistance") are more likely to be retweeted than those with neutral or factual tones.” To further address this, we have added a detailed case study tracking the iterative evolution of the rule set in Appendix B of the revised manuscript.

---

### Meta-Review · Area_Chair_Btan · 2026-01-07

**Summary:**

The paper addresses a problem of building a rule-based system using Large Language Models (LLMs). The Authors propose to generate rules using LLMs, estimate their weights using logistic regression, iteratively refine them on difficult examples, and then to assess performance of different inference strategies, such as direct inference via the trained logistic regression model or indirect inference via LLMs.

The reviewers find the approach interesting, but rather not good enough for a top-tier conference. All initial scores are negative. The most critical comments concern incremental contribution of the work, the generality of the approach (the use of the approach with other LLMs), limited and potentially flawed experimental studies, or the fact that the proposed model performs worse than a simple baseline on two datasets. The Authors prepared a rebuttal which clarifies many of the about issues. Unfortunately, as AC, I also find the paper in the current state not mature enough for publication. For example, the Authors admitted in the rebuttal that the inferior performance mentioned above was caused by the LLM Safety Refusals. As the bypass they implemented a robust retry mechanism that increases the number of retries from 1 to 5. It is not clear whether this mechanism is general enough and what other surprises can be found in the current approach. Therefore, the paper needs to go through a major revision before considering it for the acceptance.

**Reviewer Concerns:**

The Authors extended their empirical studies and clarified minor points raised by the Reviewers. Nevertheless, the contribution of the paper is still limited and rather shallow. The empirical studies can be further extended to use other LLMs and competitors.

**Reviewer Scores:**

The scores would be slightly raised, but they would not cross the bar.

---

### Decision · Program_Chairs · 2026-01-26

Reject